# Association between antenatal corticosteroid use and perinatal mortality among preterm births in hospitals in Tanzania

Stanley Mwita[1]*, Eveline Konje[2], Benjamin Kamala[3,4], Angelina Izina[5], Semvua Kilonzo[6], Andrew Kigombola[7], Karol J. Marwa[8], Mary Jande[8], Deborah Dewey[9]

1 School of Pharmacy, Catholic University of Health and Allied Sciences, Mwanza, Tanzania, 2 School of Public Health, Catholic University of Health and Allied Sciences, Mwanza, Tanzania, 3 Department of Epidemiology and Biostatistics, Muhimbili University of Health and Allied Sciences, Dar es Salaam, Tanzania, 4 Department of Research, Haydom Lutheran Hospital, Haydom, Manyara, Tanzania, 5 Department of Radiology, Bugando Medical Centre, Mwanza, Tanzania, 6 Department of Internal Medicine, Catholic University of Health and Allied Sciences, Mwanza, Tanzania, 7 Afya Kamilifu Project, Maryland global initiative, Dar es Salaam, Tanzania, 8 Department of Pharmacology, Catholic University of Health and Allied Sciences, Mwanza, Tanzania, 9 Owerko Centre at the Alberta Children's Hospital Research Institute and Departments of Pediatrics and Community Health Sciences, University of Calgary, Calgary, Canada

* stanleymwita@gmail.com

**Data Availability Statement:** All relevant data are within the manuscript and its Supporting Information files.

## Abstract

### Objectives

The primary aims of this study were to investigate if exposure to antenatal corticosteroids (ACS) was associated with lower rates of perinatal mortality (primary outcome) and other adverse perinatal outcomes (i.e., stillbirth, early neonatal mortality, APGAR score of < 7 at 5 mins, neonatal sepsis and respiratory distress syndrome) in preterm infants in hospitals in Tanzania. We also examine factors associated with administration of ACS among women at risk of preterm delivery.

### Methods

A hospital-based prospective chart review study was undertaken in four hospitals located in Nyamagana and Sengerema districts, Tanzania. The study population included all stillborn and live born preterm infants delivered between 24 to 34 weeks of gestation between July 2019 to February 2020. A total 1125 preterm infants were delivered by 1008 women (895 singletons, 230 multiple). Sociodemographic and medical data were recorded from participants' medical records.

### Results

Three hundred and fifty-six (35.3%) women were administered at least one dose of ACS between 24 to 34 weeks' gestation and 385 (34.2%) infants were exposed to ACS. Infants exposed to ACS had a lower rate of perinatal mortality (13.77%) compared to those who were not exposed (28.38%). Multivariate analysis indicated that infants exposed to ACS were less likely to die during perinatal period, aRR 0.34 (95%CI 0.26-0.44). Only one-third of the sample was provided with ACS. Administration of ACS was associated with maternal

**Funding:** The author(s) received no specific funding for this work.

**Competing interests:** The authors have declared that no competing interests exist

education, attending antenatal care more than 3 times, method used to assess gestational age, maternal infection, exposure to maternal antibiotics, delivery mode and level of health facility.

## Conclusion

ACS significantly reduced the risk in perinatal mortality among infants born preterm in a limited resource setting. However, only about one-third of eligible women were provided with ACS, indicating low usage of ACS. Numerous factors were associated with low usage of ACS in this setting.

## Introduction

Perinatal mortality refers to stillbirths and neonatal death during the first week of life. Preterm birth is one of the leading causes of perinatal mortality [1, 2]. Globally, stillbirths and early neonatal mortality remain remarkably high [3, 4]. In Tanzania, in 2018, approximately 45,000 deaths occurred during the neonatal period, which accounted for 40% of under-five deaths [5]. In 2015, ten countries accounted for 65% of stillbirths globally. Tanzania ranked 9th among these countries [6].

According to a recent report of World Health Organization, which examined the causes of death in children from 2000-2016, the main causes of death in under-five children were preterm birth complications (18%), pneumonia (16%), intrapartum-related events (12%), diarrhea (8%), neonatal sepsis (7%) and malaria (5%) [7]. More than 80% of all newborn deaths resulted from three preventable and treatable conditions – complications due to prematurity, intrapartum-related deaths (including birth asphyxia) and neonatal infections [8]. Reducing complications due to prematurity could significantly impact perinatal mortality.

Respiratory distress syndrome (RDS) is a serious complication of preterm birth and is the main cause of perinatal mortality [9, 10]. RDS is caused by the absence or insufficient production of pulmonary surfactant and the associated immaturity of the lungs [11]. Antenatal corticosteroids (ACS) are recommended by the World Health Organization (WHO) for the prevention of RDS [12] and worldwide, the use of ACS is regarded as an effective intervention for reducing preterm-associated perinatal mortality. ACS trigger the lung maturation process leading to the release of surfactant into the alveoli of the fetal lung [13]. Dexamethasone and betamethasone are corticosteroids used for this purpose. Treatment consists of 2 doses of 12 mg of betamethasone every 24 hours or 4 doses of 6 mg of dexamethasone every 12 hours; a single course of ACS is recommended for pregnant women between 24 weeks and 34 weeks of gestation who are at risk of preterm delivery within 7 days [14].

Evidence from a 2017 Cochrane review supported the use of a single course of ACS to accelerate fetal lung maturation in women at risk of preterm birth [15]. The use of ACS was associated with significant reductions in infant mortality and morbidity. However, as noted by the authors, most of the studies included in this review were conducted in high income countries in hospital settings and the results may not be applicable to low resource settings with high rates of infections. A large cluster randomized clinical trial, the ACT trial, assessed the feasibility, effectiveness, and safety of a multifaceted intervention designed to increase the use of ACS at all levels of health care in low-income and middle-income countries [15]. This study was conducted in rural and semi-urban settings in six low-and-middle income countries (Argentina, Guatemala, India, Kenya, Pakistan, and Zambia). Results revealed that despite increased

use of ACS in low-birthweight infants in the intervention groups, neonatal mortality did not decrease in these groups, and mortality increased in the population overall. Further, the risk of maternal infection appeared to have increased. In contrast, in a recently published multi-country (i.e., Bangladesh, India, Kenya, Nigeria Pakistan) randomized trial involving pregnant women between 26 weeks 0 days and 33 weeks 6 days of gestation who were at risk for preterm birth, the use of dexamethasone resulted in significantly lower risks of neonatal death alone and stillbirth or neonatal death than the use of placebo, without increasing the incidence of maternal infection [16]. However, dexamethasone was not associated with a reduction in the risk of severe RDS. A recently published Cochrane review examined the benefits and harms of optimizing ACS use for anticipated preterm birth. The study concluded that in low-resource settings, a strategy of actively promoting the use of ACS in women at risk of preterm birth may increase ACS use, but it may also carry a substantial risk of unnecessary exposure of ACS to women in whom ACS are not indicated [17]. At the population level, these effects could be associated with increased risks of stillbirth, perinatal death, neonatal death before 28 days, and maternal infection. As a result, the authors suggested that a more conservative approach to the use of ACS, particularly in low-resource settings, needs to be considered. Such an approach should take into account the efficacy of ACS when used correctly and the potential adverse effects when certain conditions are not met. The findings of these studies support the need for further research on the use and effectiveness of ACS for mothers at risk of premature delivery in hospitals in low-resource settings that have minimal resources for determining the appropriateness of ACS use.

ACS are not widely used in hospital settings in many low- and middle- income countries [18]. It has been reported that only 10% to 68% of eligible women receive ACS [19–21]. Unavailability of ACS in health care settings, inadequate prescription of ACS and the arrival of pregnant women at health care facilities in well-established labor could contribute to low use of ACS in these setting [21].

To the best of our knowledge, no studies have been conducted in Tanzania that examined factors associated with the use of ACS and the association between ACS use and perinatal mortality among preterm births. This study investigated if exposure to ACS was associated with lower rates of perinatal mortality (primary outcome) and other perinatal outcomes (stillbirth, early neonatal mortality, APGAR score of < 7 at 5 mins, neonatal sepsis and RDS) in preterm infants. We also investigated factors associated with administration of ACS to women at risk preterm deliveries in district, regional and tertiary hospitals in Nyamagana and Sengerema districts in Tanzania.

## Materials and methods

### Study design

A hospital-based prospective chart review study was conducted between July 2019 to February 2020 in four select hospitals located in Nyamagana and Sengerema districts, Tanzania, namely, Bugando Medical Centre (tertiary consultant zonal referral hospital), Sekou Toure Regional Referral Hospital, Nyamagana District Hospital and Sengerema District Designated Hospital. These hospitals provide obstetric and neonatal care services to large populations within the Lake zone in Tanzania. Nyamagana and Sengerema districts are two of the seven districts of the Mwanza region, northwest Tanzania. Total deliveries for Nyamagana and Sengerema districts in 2014 were estimated to be 25,977 and 18,718, respectively [22].

The study population included all stillborn and live born preterm infants delivered between 24 weeks 0 days to 34 weeks 6 days of gestation between July 2019 to February 2020. We excluded preterm infants born with congenital malformations. Gestational age was

determined based on women's self-reports of their last normal menstrual period, fundal height and/or ultrasound. For this study, stillbirth was defined as the death or loss of baby after 20 weeks of pregnancy that occurred before or during delivery. The baby had an Apgar score of 0 at both 1 and 5 minutes after delivery. Early neonatal mortality was defined as death of a live born neonate between zero and seven days after birth. Infants (still born and live born) were classified into one of two groups, those whose mothers received ACS and those whose mothers did not receive ACS. All women in this study who received ACS delivered within 7 days of the first dose.

Information on demographic characteristics, use of ACS and associated factors, and perinatal outcomes prior to discharge from the hospital was obtained from the medical records. Data on women and their infants from the point at which pregnant women were admitted to the hospital to their discharge was recorded. The medical charts were reviewed by the principal investigator and two research assistants who were enrolled/registered nurses working in the labor wards and neonatal units of each of the hospitals.

The primary outcome, perinatal mortality, was defined as stillbirth or early neonatal mortality before day 7. Secondary outcomes were stillbirth, early neonatal mortality, APGAR score at 5 mins, neonatal sepsis and RDS. Based on reports from previous studies [21, 23, 24], the following baseline demographics and associated factors were examined: multiple pregnancy, birth weight (grams), gestational age, neonate sex, mode of delivery, maternal antibiotics, neonatal antibiotics, level of health facility, parity, antenatal care attendance days, maternal infection and fetal heart rate. Maternal antibiotics were Penicillin and/or Ampicillin, Neonatal antibiotics were Ampicillin (50 mg/kg every 12 hours) and Gentamicin (4 mg/kg every 24 hours).

The sample size was determined using Open-Source Epidemiologic Statistics for Public Health (Open Epi) [20]. Based on a power of 95%, it was determined that a minimum sample size of 1010 infants would allow us to address our primary research aim.

## Statistical analysis

The data were analyzed using STATA Version 13. Chi-square tests were conducted on the following maternal baseline characteristics: parity, marital status, education, antenatal care attendance days, mode of delivery, method used to assess gestational age, maternal infection, maternal antibiotics and level of health facility. There analyses were used to investigate differences between women who received ACS and those who did not. Differences in perinatal outcomes between infants exposed to ACS and those who were not were examined using fishers exact test and chi-square tests where appropriate. T-tests were used to determine if there were differences in mean maternal age, mean gestational age and birth weight. Modified Poisson regressions were used to investigate the associations between ACS exposure and perinatal outcomes, as well as factors associated with the administration of ACS. Multivariate regression analyses were performed to examine the effects of administration of ACS on perinatal outcomes, controlling for factors with significant associations (gestational age, birth weight, level of health facility, multiple pregnancy and delivery mode). P-values of less than 0.05 were considered statistically significant. Data are presented as frequencies (percentages), means (standard deviations) and relative risks with 95% confidence intervals as appropriate.

## Ethics

This prospective chart review study was approved by The Catholic University of Health and Allied Sciences and Bugando Medical Centre's Joint Ethics and Research Review Committee (IRB approval No: CREC/368/2019). Secondary data were collected from medical records. No

patients were contacted for this study. To ensure confidentiality, all data were anonymized before being accessed by the study team. Medical records were accessed between July 2019 and February 2020. The ethics committee waived the need for participant informed consent.

## Results

Over an eight-month period, 1125 preterm infants were delivered to 1008 women (895 singletons, 230 twins). Three hundred and fifty-six women (35.3%) delivered within 7 days after receiving at least one dose of ACS, which resulted in 385 (34.2%) infants who were exposed to ACS between 24- and 34-weeks' gestation.

No significant differences in marital status, parity or mean maternal age were found between women who received ACS and those who did not. However, the groups differ on the following variables: maternal education, antenatal care attendance days, gestational age, method used to assess gestational age, multiple pregnancy, mode of delivery (i.e., normal vaginal, assisted vaginal, c-section), maternal antibiotics, maternal infections, and level of health facility where they delivered. A higher proportion of women who received ACS were singleton pregnancies, had not been prescribed maternal antibiotics, evidenced fewer maternal infections prior to the delivery, had a normal vagina delivery and had a secondary education. Also, a higher proportion had gestational age assessed by ultrasound and the mean gestational age of their infants was higher. Among women who received ACS, a higher proportion delivered at the tertiary zonal referral hospital. Finally, a higher proportion of women who were not administered ACS had made less than 4 visits to antenatal care clinics. (Table 1).

### The association between ACS exposure and perinatal outcomes

The overall prevalence of perinatal mortality was 23.38%. Compared with unexposed infants, those who were exposed to ACS in utero had a lower rate of perinatal mortality (13.77% vs 28.38%), stillbirth (2.08% versus 18.24%), APGAR scores of less than 7 at 5 minute (7.01% vs 25.81%), neonatal sepsis (10.08% vs 16.20%) and RDS (17.92% vs 21.49%). However, early neonatal mortality was similar between exposed (11.94%) and unexposed (12.40%) infants. Univariate analyses revealed significant differences between infants exposed to ACS and those not exposed for perinatal mortality, stillbirth, APGAR score of < 7 at 5 minutes, neonatal sepsis, and RDS (Table 2).

Adjusted multivariate analysis revealed that exposure to ACS was significantly associated with a lower likelihood of perinatal mortality, aRR 0.34 (95%CI 0.26 -0.44), stillbirth, aRR 0.07 (95%CI 0.03 -0.14), early neonatal mortality, aRR 0.59 (95%CI 0.41 -0.84), APGAR scores of < 7 at 5-minute, aRR 0.20 (95%CI 0.13 -0.29), diagnoses of neonatal sepsis, aRR 0.57 (95% CI 0.41 -0.80), and RDS, aRR 0.58 (95%CI 0.44-0.78) (Table 3).

### Factors associated with administration of ACS

Adjusted multivariate analysis revealed that administration of ACS was associated with maternal education, antenatal care attendance days, delivery mode, exposure to maternal antibiotics, maternal infection, level of health facility and method used to assess gestational age. Women who received ACS were more likely to have had secondary education compared to those without formal education, aRR 1.83 (95%CI 1.21-2.72), attended antenatal care for at least 4 visits, aRR 1.21 (95%CI 1.02-1.42), and delivered via C-section compared to a normal vaginal delivery, aRR 1.25 (95%CI 1.09-1.44). They were also less likely to have been prescribed maternal antibiotics aRR, 0.68 (95%CI 0.60-0.79), evidenced lower levels of maternal infection, aRR 1.55 (95%CI 1.14-2.11) and delivered at the tertiary zonal referral hospital compared to a district hospital aRR 8.70 (95% CI 5.06-14.98). Further, women who received ACS were less likely to

**Table 1. Maternal demographic, medical and health facility characteristics by ACS treatment.**

| | ACS (n=356) | No ACS (n=652) | p Value |
|---|---|---|---|
| **Maternal Demographics** | M (SD)/N (%) | M (SD)/N (%) | |
| Mean maternal age (years) | 26.8 ± 5.9 | 26.5 ± 6.3 | 0.522 |
| Parity | | | |
| Nulliparous | 128 (35.96) | 195 (29.91) | 0.050 |
| Parous | 228 (64.04) | 457 (70.09) | |
| Marital Status | | | |
| Married | 311 (87.36) | 562 (86.20) | 0.604 |
| Single | 45 (12.64) | 90 (13.80) | |
| Education | | | |
| College and above | 55 (15.45) | 51 (7.82) | <0.001 |
| Secondary education | 178 (50) | 205 (31.44) | |
| Primary education | 105 (29.49) | 343 (52.61) | |
| No formal education | 18 (5.06) | 53 (8.13) | |
| **Medical Variables** | | | |
| Antenatal care visits* | | | |
| ≥ 4 | 160 (45.98) | 214 (34.91) | 0.001 |
| 1-3 | 188 (54.02) | 399 (65.09) | |
| Mean gestational age (weeks) | 31.9 ±2.2 | 31.3 ± 2.4 | <0.001 |
| Method used to assess gestational age | | | |
| Maternal self-report of the last normal menstrual period | 107 (30.06) | 437 (67.02) | <0.001 |
| Ultrasound | 223 (62.64) | 141 (21.63) | |
| Fundal height | 26 (7.30) | 74 (11.35) | |
| Mode of delivery | | | |
| Assisted vaginal | 13 (3.65) | 17 (2.61) | <0.001 |
| C- section | 156 (43.82) | 118 (18.10) | |
| Normal vaginal | 187 (52.53) | 517 (79.29) | |
| Multiple pregnancy | | | |
| No | 328 (92.13) | 567 (86.96) | 0.013 |
| Yes | 28 (7.87) | 85 (13.04) | |
| Maternal infection | | | |
| No | 327 (91.85) | 558 (85.58) | 0.004 |
| Yes | 29 (8.15) | 94 (14.42) | |
| Maternal antibiotics | | | |
| No | 207 (58.15) | 489 (75) | <0.001 |
| Yes | 149 (41.85) | 163 (25) | |
| Level of health facility | | | |
| Tertiary zonal hospital | 256 (71.91) | 133 (20.40) | <0.001 |
| Regional hospital | 85 (23.88) | 261 (40.03) | |
| District hospital | 15 (4.21) | 258 (39.57) | |

*Denominator included only those who attended antenatal care (ACS, n=348) & (No ACS, n=613).

have had gestational age assessed by maternal self-report of the last normal menstrual period compared to ultrasound, aRR 0.55 (95%CI 0.46-0.66) (Table 4). The rates of ACS administration did not differ between women with singleton pregnancies and those with multiple pregnancies, aRR1.18 (95%CI0.88-1.58).

Examination of the reasons for not administering ACS revealed that more than half of the women who did not receive ACS arrived at the hospital in well-established labor. Being out of

**Table 2. Perinatal outcomes based on ACS exposure.**

| | ACS (n=385) | Not ACS (n=740) | *p* Value |
|---|---|---|---|
| Outcomes | n (%) | n (%) | |
| Perinatal mortality | 53 (13.77) | 210 (28.38) | <0.001 |
| Stillbirth | 8 (2.08) | 135 (18.24) | <0.001 |
| Early neonatal mortality | 45 (11.94) | 75 (12.40) | 0.830 |
| APGAR score <7 at 5 minutes | 27 (7.01) | 191(25.81) | <0.001 |
| Neonatal sepsis | 38 (10.08) | 98 (16.20) | 0.007 |
| RDS | 69 (17.92%) | 159 (21.49) | 0.004 |

stock was indicated as the reason for not administering ACS for only 4% of the participants. For 17% of the participants, no reason was provided by prescribers for not prescribing ACS (Fig 1).

## Discussion

In this study, the overall prevalence of perinatal mortality was 23.38% among preterm deliveries between 24 to 34 weeks. Preterm infants exposed to ACS in utero in this low resource country, were less likely to evidence perinatal mortality. This finding is consistent with a recently updated Cochrane review of randomized controlled trials conducted in hospital settings, which reported a lower rate of perinatal mortality regardless of resource setting among infants who were exposed to ACS (RR 0.85, 95% CI 0.77 -0.93; 14 studies, 9833 infants) [25]. Our findings, however, were not consistent with those of the ACT trial, which reported no difference in perinatal mortality between the ACS exposed and unexposed groups [15]. This could be attributable to differences in health care settings and differences in maternal and perinatal baseline characteristics. The ACT trial reported on deliveries from all levels of care, including primary health care and care at the community level. In contrast, the present study reported on only hospital deliveries at district, regional and zonal hospitals.

The WHO recommends ACS for women at risk of preterm birth from 24 weeks to 34 weeks' gestation in settings where the following criteria are met: gestational age assessment can be accurately undertaken, preterm birth is considered imminent, there is no clinical evidence of maternal infection, adequate childbirth care is available and the preterm newborn can receive adequate care if needed (including resuscitation, thermal care, feeding support, infection, treatment and safe oxygen use) [12]. In the present study, administration of ACS was associated with not only a reduced risk of perinatal mortality but also a reduced risk of

**Table 3. Univariate and multivariate analysis of the association between ACS exposure and perinatal outcomes.**

| Outcomes | Crude relative risk (95%CI) | *Adjusted relative risk (95%CI) |
|---|---|---|
| Perinatal mortality | 0.48 (0.37-0.64) | 0.34 (0.26-0.44) |
| Stillbirth | 0.11 (0.06-0.23) | 0.07 (0.03-0.14) |
| Early neonatal mortality | 0.96 (0.68-1.36) | 0.59 (0.41-0.84) |
| APGAR scores <7 at 5 minutes | 0.27 (0.19-0.40) | 0.20 (0.13-0.29) |
| Neonatal sepsis | 0.62 (0.44-0.88) | 0.57 (0.41-0.80) |
| RDS | 0.70 (0.54-0.89) | 0.58 (0.44-0.78) |

*Adjusted for gestational age, birth weight, level of health facility, multiple pregnancy, delivery mode, while neonate sex, maternal antibiotics, neonatal antibiotics, parity, antenatal care attendance days, maternal infection and fetal heart rate were not significant hence removed from the model.

**Table 4. Univariate and multivariate analysis of the factors associated with administration of ACS.**

| Factors | | Crude relative risk (95%CI) | Adjusted relative risk (95%CI) |
|---|---|---|---|
| Education | College and above | 2.05 (1.32-3.18) | 1.71 (1.13-2.60) |
| | Secondary education | 1.83 (1.21-2.77) | 1.83 (1.21-2.72) |
| | Primary education | 0.92 (0.60- 1.42) | 1.76 (1.16-2.69) |
| | No formal education | 1 | 1 |
| Antenatal care visits | ≥ 4 | 1.34 (1.10-1.64) | 1.21 (1.02-1.42) |
| | 1-3 | 1 | 1 |
| Delivery mode | Assisted delivery | 1.63 (1.06 - 2.50) | 1.54 (1.06-1.99) |
| | C-section | 2.14 (1.82 - 2.52) | 1.25 (1.09-1.44) |
| | Normal vaginal | 1 | 1 |
| Maternal antibiotics | No | 0.62 (0.53-0.73) | 0.68 (0.60-0.79) |
| | Yes | 1 | 1 |
| Maternal infection | No | 1.57 (1.13-2.18) | 1.55 (1.14-2.11) |
| | Yes | 1 | 1 |
| Level of health facility | Tertiary Zonal Hospital Referral Hospital | 11.98 (7.28-19.69) | 8.70 (5.06-14.98) |
| | Regional | 4.47 (2.64-7.56) | 3.68 (2.13-6.35) |
| | District | 1 | 1 |
| Method used to assess gestational age | Maternal self-report of the last normal menstrual period | 0.32 (0.26-0.39) | 0.55 (0.46-0.66) |
| | Fundal height | 0.42 (0.30-0.60) | 0.76 (0.55-1.05) |
| | Ultrasound | 1 | 1 |
| Multiple pregnancy | No | 1.58 (1.06-2.06) | 1.18 (0.88-1.58) |
| | Yes | 1 | 1 |

other adverse perinatal outcomes including stillbirth, early neonatal mortality, neonatal sepsis, RDS and APGAR score of < 7 at 5 min. In contrast, the recently published WHO ACTION trial reported that administration of the ACS to women at risk for early preterm birth in low-resource countries resulted in a significantly lower risk of early neonatal death but no difference in risk of stillbirth, neonatal sepsis, severe RDS at 24 hours after birth or APGAR scores of <7 at 5 min after birth [16]. Differences in the findings between the present study and the WHO ACTION Trial could be due to differences in methodology, study setting and participant characteristics. For example, the WHO ACTION trial was a randomised control trial conducted in 29 secondary- and tertiary-level hospitals across Bangladesh, India, Kenya, Nigeria, and Pakistan, whereas the present study was a prospective observational chart review study conducted only in Tanzania. The present study included clinical settings with diverse neonatal care resources, from primary to tertiary level hospitals, that may have had differing methods and criteria for assessing eligibility for ACS treatment; gestational age was determined by either maternal self-reports of last menstrual period, measurement of fundal height or ultrasonographic examination. In contrast, the WHO ACTION trial was undertaken in hospital settings where the WHO criteria for ACS treatment could be met; gestational age determined by ultrasonographic examination. Also, in the WHO ACTION trial, women with maternal infection were excluded; however, in the present study, 8.15% of women who were administered ACS had maternal infection.

In the present study of women who delivered between 24- and 34-weeks' gestation, only 35.3% were administered ACS. This is consistent with a study conducted in Ecuador [26]. In contrast, some studies conducted in low- and middle-income countries have also reported low rates of ACS use, e.g in Malawi (10%), Democratic Republican of Congo (16%), Nepal (20%), Uganda (27%), Nigeria (30%) and Kenya (32%) [27, 28], while others conducted in South East

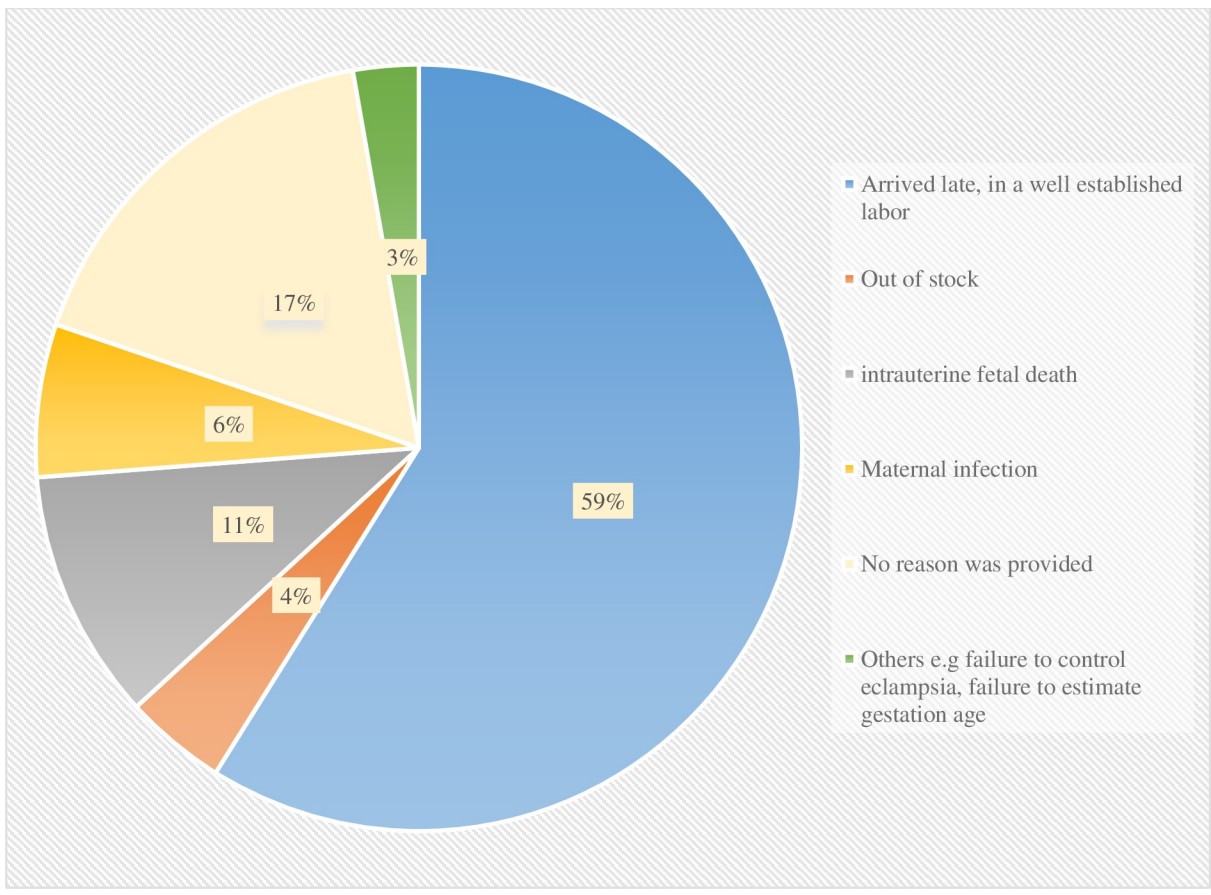

**Fig 1. Reasons for not administering ACS.**

Asia have reported a higher use of ACS (40%) [29]. It is notable that, in our study, more than half of the women were not administered ACS because they arrived at the hospital in a well-established labor. This suggests that more efforts need to be directed at educating women on risk of preterm delivery and early signs of labor and they should be encouraged to seek medical advice and visit health facilities early if they experience any signs of labor. Low usage of ACS in low resource settings has also been attributed to lack of standard treatment guidelines, prescribers' attitudes, awareness or skills, patient access to proper health facilities and availability of ACS [30]. Availability of ACS was not found to be a significant contributor to non-administration of ACS in the present study and was reported in only 4% of the sample. However, 17% of physicians did not provide a reason for not prescribing ACS. This could be due to lack of knowledge of the benefit of ACS in reducing perinatal mortality or other unidentified factors.

In our study, pregnant women who delivered at a tertiary zonal hospital, were more likely to receive ACS than women who delivered at district hospitals. This could be attributed to the higher level of care provided at this facility, the level of knowledge of the medical professionals, and the availability of obstetricians and gynecologists in this setting. Increasing health care professionals' awareness and knowledge on the use and importance of ACS in reducing perinatal mortality is a priority.

The strengths of the present study were its use of a prospective design and a large sample. Moreover, the current study is unique as we investigated potential reasons for the non-administration of ACS, as well as factors associated with their administration. However, our study

has some limitations. First, it was a prospective observational study. Thus, the inherent biases associated with observational studies (e.g., selection bias, information bias, confounding) could have influenced our outcomes. We attempted to control for selection bias by including all stillbirths and live births of preterm infants delivered at the selected hospitals between 24 weeks 0 days to 34 weeks 6 days of gestation between July 2019 to February 2020. Informational bias was controlled for by using a standard data collection form and confounding by considering a number of factors that could be associated with exposure to ACS and pregnancy outcomes. A second limitation is that we did not explicitly consider indications for preterm delivery (e.g., preterm labor, pre-eclampsia, premature preterm rupture of membranes and placenta previa) on perinatal mortality. Third, we did not consider the influence of small for gestation age to the perinatal outcomes.

## Conclusion

ACS significantly reduced the risk in perinatal mortality among infants born preterm in Tanzania, a low resource country. Administration of ACS also reduced the risk of other adverse perinatal outcomes including stillbirth, early neonatal mortality, APGAR score of $< 7$ at 5 min, neonatal sepsis and RDS. However, only about one-third of eligible women were provided with ACS, indicating low usage of ACS in this setting. Health care professionals, particularly in lower-level facilities, need to be educated on the importance of ACS in reducing perinatal mortality among preterm infants and the WHO criteria for their administration. Also, women at risk of preterm birth need to be identified early by health care professionals. Finally, antenatal care clinics need to education women about the signs of preterm birth and encouraged them to attend a hospital if they experience such symptoms. Such steps could significantly reduce the rates of perinatal mortality among infants born preterm in Tanzania.

## Supporting information

**S1 Data.**
(XLSX)

## Acknowledgments

We wish to acknowledge the invaluable support of Bugando Medical Centre director and medical officers in-charge of Sekou Toure Regional Referral Hospital, Nyamagana District Hospital and Sengerema District Designated Hospital.

## Author Contributions

**Conceptualization:** Stanley Mwita, Benjamin Kamala, Deborah Dewey.

**Data curation:** Stanley Mwita, Angelina Izina, Semvua Kilonzo, Andrew Kigombola, Karol J. Marwa.

**Formal analysis:** Stanley Mwita, Eveline Konje.

**Supervision:** Mary Jande, Deborah Dewey.

**Writing – original draft:** Stanley Mwita, Angelina Izina.

**Writing – review & editing:** Eveline Konje, Benjamin Kamala, Semvua Kilonzo, Andrew Kigombola, Karol J. Marwa, Mary Jande, Deborah Dewey.

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
