## [Decision Letter · Decision Letter 0]

18 Mar 2021

PONE-D-21-04378

Association between antenatal corticosteroid use and perinatal mortality among preterm births in hospitals in Tanzania.

PLOS ONE

Dear Dr. Mwita,

Thank you for submitting your manuscript to PLOS ONE. After careful consideration, we feel that it has merit but does not fully meet PLOS ONE’s publication criteria as it currently stands. Therefore, we invite you to submit a revised version of the manuscript that addresses the points raised during the review process.

We look forward to receiving your revised manuscript.

Kind regards,

Andrew Sharp, PhD

Academic Editor

PLOS ONE

Journal Requirements:

2. In the ethics statement in the manuscript and in the online submission form, please provide additional information about the patient records used in your retrospective study, including: a) whether all data were fully anonymized before you accessed them; b) the date range (month and year) during which patients' medical records were accessed; c) the date range (month and year) during which patients whose medical records were selected for this study sought treatment. If the ethics committee waived the need for informed consent, or patients provided informed written consent to have data from their medical records used in research, please include this information.

Reviewers' comments:

Reviewer's Responses to Questions

**Comments to the Author**

1. Is the manuscript technically sound, and do the data support the conclusions?

Reviewer #1: Partly

2. Has the statistical analysis been performed appropriately and rigorously? 

Reviewer #1: Yes

3. Have the authors made all data underlying the findings in their manuscript fully available?

Reviewer #1: Yes

4. Is the manuscript presented in an intelligible fashion and written in standard English?

Reviewer #1: Yes

5. Review Comments to the Author

Reviewer #1: Thank you for the opportunity to read this manuscript which is certainly very topical at the minute. Overall the manuscript is well presented. However I have a number of queries and suggestions detailed below.

BACKGROUND:

I would suggest the authors update the background to strengthen the rationale for their cohort study. I would discuss the Althabe study, the ACTION trial and the updated Cochrane antenatal corticosteroid review in more detail (reference 22 has been updated).

In addition the authors have not mentioned the Cochrane review entitled; Strategies for optimising antenatal corticosteroid administration for women with anticipated preterm birth which is extremely pertinent to their trial.

METHODS:

Line 107: The authors definition of stillbirth. Is this not neonatal death?

Line 127: I would rephrase 'explanatory variables' perhaps to baseline demographics and associated factors

Lines 127-128: ACS should not be included among the explanatory variables

Data analysis:

Line 142: The authors mention apgar scores and neonatal sepsis-this is not covered in the initial aims outlined in the study. There needs to be consistency.

The authors need to distinguish between study variables and demographics of the population and consider this when undertaking analysis about antenatal corticosteroid use. The authors need to be careful that any analysis undertaken is clinically meaningful/relevant rather than data mining.

Results:

Line 209: The authors primary aim relates to perinatal mortality-the authors report on apgar scores, neonatal sepsis and RDS. This is not specified in the initial methods.

I think the authors should review the Tables-Table 3 and Table 4 are not particularly reader friendly

Line 259: Gestational age should not be included in this table

Discussion:

Overall I think the discussion section need to be strengthened.

The authors also need to discuss the WHO recommended criteria prior to antenatal corticosteroid administration. As one of the criterion is that gestational age assessment needs to be accurately undertaken and that there is no clinical evidence of maternal infection. I would imagine this might impact on administration of antenatal corticosteroids so should be discussed as an explanation for their results.

Line 277/278: This cochrane review has been updated and includes the action study. This paragraph needs to be changed and rephrased to reflect the new evidence/conclusions.

Line 282: I do not think the references used are the most appropriate given the recently published studies.

Lines 295/296: I think this needs to be rephrased the ACTION trial was a randomised control trial including over 2852 women and 3070 fetuses. The authors need to be a little careful not to over reach and there should be some mention regarding the limitations of their study as it is not a randomised trial and the inherent biases associated with cohort studies.

There are a few typographical errors.

Lines: 250, 300

6. PLOS authors have the option to publish the peer review history of their article (what does this mean?). If published, this will include your full peer review and any attached files.

Reviewer #1: **Yes: **Emma McGoldrick

---

## [Author Response · Author response to Decision Letter 0]

23 Apr 2021

We have rephrased the ethics statement to include additional information as requested. Also, we have responded to each of the reviewers’ comments.

---

## [Editor Report · Decision Letter 1]

7 Jul 2021

Association between antenatal corticosteroid use and perinatal mortality among preterm births in hospitals in Tanzania.

PONE-D-21-04378R1

Dear Dr. Mwita,

We’re pleased to inform you that your manuscript has been judged scientifically suitable for publication and will be formally accepted for publication once it meets all outstanding technical requirements.

Kind regards,

Andrew Sharp, PhD

Academic Editor

PLOS ONE

Additional Editor Comments (optional):

many thanks for making the required changes to the manuscript submission
---

## [Editor Report · Acceptance letter]

13 Jul 2021

PONE-D-21-04378R1 

Association between antenatal corticosteroid use and perinatal mortality among preterm births in hospitals in Tanzania 

Dear Dr. Mwita:

I'm pleased to inform you that your manuscript has been deemed suitable for publication in PLOS ONE. Congratulations! Your manuscript is now with our production department. 

Kind regards, 

on behalf of

Dr. Andrew Sharp 

Academic Editor

PLOS ONE